# Hormonal Defects Are Common during Puumala Hantavirus Infection and Associate with Disease Severity and Biomarkers of Altered Haemostasis

**DOI:** 10.3390/v13091818

**Published:** 2021-09-13

**Authors:** Marlene Tarvainen, Satu Mäkelä, Outi Laine, Ilkka Pörsti, Sari Risku, Onni Niemelä, Jukka Mustonen, Pia Jaatinen

**Affiliations:** 1Department of Internal Medicine, Tampere University Hospital, 33520 Tampere, Finland; satu.m.makela@pshp.fi (S.M.); outi.laine@pshp.fi (O.L.); ilkka.porsti@tuni.fi (I.P.); jukka.mustonen@tuni.fi (J.M.); pia.jaatinen@tuni.fi (P.J.); 2Faculty of Medicine and Health Technology, Tampere University, 33520 Tampere, Finland; 3Division of Internal Medicine, Seinäjoki Central Hospital, 60220 Seinäjoki, Finland; sari.risku@epshp.fi; 4Laboratory and Medical Research Unit, Seinäjoki Central Hospital, 60220 Seinäjoki, Finland; onni.niemela@epshp.fi

**Keywords:** hantavirus, puumala virus, HFRS, hormonal defect, AKI, hypogonadism, hypothyroidism, coagulation, fibrinolysis

## Abstract

Central and peripheral hormone deficiencies have been documented during and after acute hantavirus infection. Thrombocytopenia and coagulation abnormalities are common findings in haemorrhagic fever with renal syndrome (HFRS). The associations between coagulation and hormonal abnormalities in HFRS have not been studied yet. Forty-two patients diagnosed with Puumala virus (PUUV) infection were examined during the acute phase and on a follow-up visit approximately one month later. Hormonal defects were common during acute PUUV infection. Overt (clinical) hypogonadism was identified in 80% of the men and approximately 20% of the patients had overt hypothyroidism. At the one-month follow-up visit, six patients had central hormone deficits. Acute peripheral hormone deficits associated with a more severe acute kidney injury (AKI), longer hospital stay and more severe thrombocytopenia. Half of the patients with bleeding symptoms had also peripheral hormonal deficiencies. Patients with free thyroxine levels below the reference range had higher D-dimer level than patients with normal thyroid function, but no thromboembolic events occurred. Acute phase hormonal abnormalities associate with severe disease and altered haemostasis in PUUV infection.

## 1. Introduction

Hantavirus infections present with two distinct clinical syndromes: a haemorrhagic fever with renal syndrome (HFRS) in Europe and Asia, and a hantavirus cardiopulmonary syndrome (HCPS) in North and South America [1]. PUUV is the most common hantavirus in Europe, causing a mild form of HFRS, also called nephropathia epidemica [1,2]. The reservoir host of PUUV is the bank vole (Myodes glareolus) [2]. In Finland the seroprevalence of PUUV infection in adult population is about 12.5% [3], and approximately 1000–3000 serologically verified infections occur annually [1,2].

The clinical course of PUUV infection is usually mild but can vary from subclinical to fatal [2]. Mortality is low, approximately 0.1%. The typical symptoms of PUUV infection include fever, headache, visual disturbances, nausea, backache, and abdominal pain. Serious haemorrhages are rare, but mild bleeding manifestations, such as conjunctival bleeding, petechiae, or epistaxis, occur in about one-third of the patients [2,4]. Transient acute kidney injury (AKI), thrombocytopenia and increased vascular permeability leading to capillary leakage are typical findings in PUUV infection.

Increased platelet consumption is a central mechanism leading to thrombocytopenia in acute HFRS [2,5]. Platelet activation, enhanced thrombin formation and fibrinolysis together with intravascular coagulation have all been documented [5,6,7]. Furthermore, high mean platelet volume (MPV), immature platelet fraction (IPF%), and serum thrombopoietin level have been reported, indicating active thrombopoiesis during thrombocytopenia in acute PUUV infection [6,7].

Case reports have been published on hypophyseal haemorrhage and panhypopituitarism during acute PUUV infection [8,9,10,11,12,13]. PUUV antigens have also been detected in the pituitary of a patient with a fatal HFRS [9]. Central and peripheral hormone deficiencies have been identified during both acute HFRS and months or even years later [8,9,10,11,12,14,15,16,17]. In a previous retrospective study [15] serum cortisol and prolactin levels were higher and serum testosterone concentrations lower during acute PUUV infection, when compared with the corresponding values measured three months later. These acute hormonal alterations were related to the severity of AKI and inflammation.

An increased tendency for bleeding has been reported among patients with severe hypothyroidism, while mainly a thrombotic tendency and a hypercoagulable state have been related to other endocrine disorders (e.g., endogenous hypercortisolism, hypogonadotropic hypogonadism, subclinical or mild hypothyroidism, hyperthyroidism, growth hormone deficiency, acromegaly, hyperprolactinaemia) [18,19]. The associations between hormonal deficiencies and platelet or coagulation abnormalities in HFRS have not been studied previously.

The first objective of the present study was to study, in a prospective setting, the hormonal abnormalities during an acute PUUV infection and at a follow-up visit one month later. The second objective was to investigate the association of the hormonal alterations with the severity of acute PUUV infection. Thirdly, we wanted to examine whether the hormonal abnormalities are related to the decreased platelet count and enhanced thrombin formation, fibrinolysis, and thrombopoiesis during acute PUUV infection.

## 2. Materials and Methods

### 2.1. Patients

The study consisted of a prospectively collected cohort of 42 consecutive patients with a serologically confirmed acute PUUV infection [20], treated at Tampere University Hospital or Seinäjoki Central Hospital, Finland, during the years 2010–2017. Thirty of them have participated in our previous study evaluating platelet formation and functions during PUUV infection [7]. Every patient gave a written informed consent before enrolments in the study and the Ethics Committee of Tampere University Hospital approved the study protocol (R09206, accessed on 10 December 2009 and R15007, accessed on 14 January 2015).

The median age of the patients was 45 years (range 21 to 67 years), and 27 (64%) of them were male. Nineteen patients had concomitant diseases, including arterial hypertension (*n* = 7), diabetes (Type 1, *n* = 1; Type 2, *n* = 3), coronary artery disease (*n* = 2), hypercholesterolaemia (*n* = 3), sleep apnoea (*n* = 2), coeliac disease (*n* = 2); and hereditary spherocytosis with splenectomy [21], hypothyroidism, gastro-oesophageal reflux disease, ulcerative colitis, spondylarthritis, chronic obstructive pulmonary disease, asthma, prostatic hyperplasia, and glaucoma one of each.

One female patient used oral contraception, four patients had a hormonal intrauterine device and one patient used postmenopausal oestrogen replacement therapy. Postmenopausal women (*n* = 6) and premenopausal women with hormonal medications (*n* = 4) were excluded from the analyses of hypogonadism. All the hormone measurements, however, were included in the comparison of hormone levels between the acute PUUV infection and one month after the acute disease. No other medications that could possibly alter plasma hormone concentrations were used during the study. None of the patients were on immunosuppressive or anticoagulation therapy. Two patients used anti-platelet therapy (acetylsalicylic acid) at the time of hospital admission, but the medication was discontinued until discharge from hospital. Otherwise, the patients were on their regular medications throughout the study.

### 2.2. Study Protocol

A detailed past and current medical history was obtained, and a careful physical examination was performed. Blood specimens were collected between 7.30 and 9.30 a.m. Laboratory samples for plasma creatinine, C-reactive protein (CRP), and blood cell counts were measured repeatedly, as clinically needed. The highest plasma creatinine concentration measured during the hospital stay reflects the severity of PUUV-induced AKI. The difference between the maximum and minimum body weight during the hospital stay (weight change) reflects the severity of AKI and the phenomenon of capillary leakage. Blood leukocyte count and plasma CRP level reflect the degree of inflammation during the acute infection, and the lowest platelet count measured illustrates the patient’s risk for bleeding. The length of hospital stay reflects the overall severity of illness.

The acute-phase blood samples for the assessment of thrombin formation, fibrinolysis and thrombopoiesis (*n* = 27), and for the hormone analyses (*n* = 40) were collected on the first morning after hospital admission, a median of 7 days (4 to 12 days) after the onset of fever. Forty patients attended the follow-up visit, approximately one month (17 to 71 days) after discharge from hospital.

### 2.3. Laboratory Measurements

The blood cell counts, plasma creatinine and CRP concentrations were measured by standard laboratory methods at Fimlab Laboratories, Tampere. Plasma D-dimer and prothrombin fragments (F1 + 2), mean platelet volume (MPV), and immature platelet fraction (IPF%), and serum thrombopoietin levels were measured as described in detail previously [5,7]. Hormone analyses were measured by accredited methods at Fimlab Laboratories, Tampere (detailed methodological data in Appendix A).

### 2.4. Statistics

The highest and the lowest values of the various variables measured during hospitalisation for each patient were designated as the maximum and minimum values, respectively. To describe the data, median, minimum, and maximum values are given for continuous variables, and numbers and percentages for categorical variables. Categorical data were analysed by the Chi-square test or the Fisher exact test, as appropriate. Continuous numerical data were analysed by the Kruskal–Wallis test, and after a statistically significant test result, post hoc comparisons between the groups were performed using the Mann–Whitney U test. The Wilcoxon signed-rank test was used for evaluating changes in serum hormone levels between the acute and the follow-up visit. The Spearman rank correlation coefficient was used for correlations between continuous variables. A two-sided *p*-value of less than 0.05 was regarded as statistically significant, and the Bonferroni correction was applied in all post hoc analyses. IBM SPSS Statistics for Windows, Version 26.0 (IBM Corporation, Armonk, NY, USA) was used for the statistical analyses.

## 3. Results

The basic clinical characteristics and laboratory findings of the 42 patients with acute PUUV infection are presented in Table 1. Fever lasted for a median of 9 days (3 to 16 days). None of the patients were diagnosed with a bacterial infection. Almost all patients (95%) presented with thrombocytopenia (platelet count below 150 × 10^9^/L), but bleeding symptoms (nasal bleeding, haemoptysis, haematochezia or melaena, petechiae) occurred in only eight patients (22%), whilst major bleeding complications were not observed. None of the patients suffered from clinical thromboembolic events.

Thirty-nine (93%) of the patients had AKI (maximum plasma creatinine level >100 µmol/L for males and >90 μmol/L for females) during the acute PUUV infection. Two patients received transient haemodialysis treatment. Clinical shock (defined as typical clinical symptoms of shock and systolic blood pressure <90 mmHg) was diagnosed in four patients. One male patient with a previous splenectomy due to hereditary spherocytosis [21] and another male patient were treated at the intensive care unit due to severe clinical course of PUUV infection. One patient suffered from a Guillain–Barré syndrome (probably induced by PUUV infection) and was treated with plasmapheresis at the stroke unit. Due to their critical illnesses, acute-phase study samples were not obtained from the patients with the spherocytosis and the Guillain–Barré syndrome. All patients recovered completely.

### 3.1. Hormonal Alterations during Acute PUUV Infection

During the acute PUUV infection, serum cortisol and prolactin (PRL) levels were significantly higher, and serum insulin-like growth factor 1 (IGF-1), follicle stimulating hormone (FSH), luteinising hormone (LH) and testosterone (Testo) levels lower than the corresponding levels at the follow-up visit one month later (Table 2). Plasma free thyroxine (fT4) concentrations were significantly lower and thyrotropin (TSH) slightly higher during the acute phase, compared with the levels at the one-month visit (Table 2).

Altogether 25 out of 40 (63%) patients had overt defects of the gonadal or thyroid axis during the acute PUUV infection. Twenty-one patients (53%) had central hormone defects, while overt peripheral deficits were observed in 6 patients (15%), and subclinical peripheral deficits in 5 patients (13%) (Figure 1). Eight patients had two different hormonal defects at the same time.

The acute hormone deficits were more common and more severe in male than in female patients. Twenty out of the 25 (80%) male patients, whose hormone levels were analysed in the acute phase, had overt peripheral (*n* = 3, 12%) or central (*n* = 17, 68%) hypogonadism, while two out of five (40%) premenopausal women without hormonal contraception had central hypogonadism. Overt thyroid axis defects (fT4 below the reference range) were seen in approximately 20% of the patients and these defects, as well, were more frequent in male than in female patients (28% of men having clinical hypothyroidism vs. 13% of women). Plasma thyroid peroxidase antibody (TPOAb) level was above the reference range in four (10%) patients, none of them having overt peripheral hypothyroidism.

Five patients had a relatively low serum cortisol level (<300 nmol/L) during the acute illness. All of these patients had also transient hyponatraemia (plasma sodium level 122–137 mmol/L) in the absence of significant hypotension (minimum blood pressure 101–135 /63–78 mmHg). The serum IGF-1 concentration was below the reference range in 10 patients (25%), and the IGF-1:GH-ratio (insulin-like growth factor 1: growth hormone-ratio) in the whole cohort was lower during the acute phase (median 12, range 0.4 to 130) than at the control visit (median 77, range 3.8 to 630).

### 3.2. Hormonal Abnormalities at the Follow-Up Visit

Overt gonadal or thyroid axis defects were still found in six of the 40 (15%) patients approximately one month after discharge from hospital. All of them had a central hormone deficit. Four out of 25 men (16%) and one premenopausal woman had central hypogonadism. This woman developed panhypopituitarism and polyendocrinopathy a few months after the acute PUUV infection [17]. Only one patient had central hypothyroidism (low fT4, normal TSH) at the one-month follow-up visit. In addition, four men (16%) had subclinical peripheral hypogonadism (normal Testo, elevated LH level), and six patients (15%) had subclinical peripheral hypothyroidism (normal fT4, elevated TSH). Plasma TPOAb concentration was above the reference range in three patients (8%), who also had had elevated TPOAbs during the acute PUUV infection. One of these patients had subclinical peripheral hypothyroidism. Two patients (male and female) (6%) had low cortisol (128–161 nmol/L) and normal ACTH levels (4–13 ng/L) at the follow-up visit, one of them having symptoms related to low cortisol level (fatigue, decreased muscle strength). An overview of the clinical and subclinical hormonal deficits during the acute PUUV infection and at the follow-up visit are presented in Figure 1.

Elevated hormone levels were observed in a few patients at the follow-up visit. Three patients (8%) had a high serum cortisol level, associated with a plasma ACTH level within the reference range. The PRL concentration was slightly elevated (344–535 mU/L) in six patients (17%). One of these patients had central hypogonadism and subclinical peripheral hypothyroidism, one had subclinical peripheral hypothyroidism, and two patients had subclinical hypogonadism. Four out of 15 women (27%) and one man had an elevated IGF-1 concentration (26–37 nmol/L).

### 3.3. Associations of Hormone Levels with the Severity of PUUV Infection

There were several significant correlations between the hormone concentrations measured in the acute phase and the markers of PUUV infection severity. The strongest correlations between the hormone levels and the markers of disease severity are illustrated in Figure 2.

During the acute phase, there were significant differences in the markers of severity of PUUV infection between the patients with overt (clinical) hormonal deficiencies and those with normal hormone levels (Table 3). Males had overt hormone defects more often than females (Table 3).

No differences were found in the age distribution, clinical picture, or basic laboratory findings during acute PUUV infection between the six patients who presented with overt hormone deficiencies at the one-month follow-up visit and those with normal hormone levels (data not shown).

### 3.4. Associations of Hormone Levels with Platelets, Thrombin Formation and Fibrinolysis

The minimum platelet counts were significantly lower and the IPF% values higher in patients with peripheral hormone deficits than in those with normal hormone levels during acute PUUV infection (Table 4). Median levels of prothrombin fragments and D-dimer were also numerically higher in patients with peripheral hormone deficits, but the difference was not statistically significant. Nevertheless, the median and maximum D-dimer levels were more than four times higher among the patients with peripheral hormonal deficiencies, compared to the patients with normal hormone levels (Table 4).

Serum cortisol concentration was the hormone level showing strongest associations with the platelet variables, as well as with the markers of thrombin formation and fibrinolysis (Figure 3). GH concentrations correlated inversely with the minimum platelet count (r = −0.468, *p* = 0.002), and the IGF-1 and fT4 levels with IPF% (r = −0.501, *p* = 0.006 and r = −0.478, *p* = 0.009, respectively).

The patients with fT4 level below the reference range had higher D-dimer levels than the patients with normal thyroid function (median 14.2 mg/L, range 3.1 to 34.0 mg/L vs. median 2.1 mg/L, range 0.8–29.6 mg/L, *p* = 0.007). In patients with hyperprolactinaemia, IPF% (11.1%, (2.5–15.1%) vs. 5.2%, (1.8–23.8%), *p* = 0.046) and MPV (11.9 fL, (10.4–13.1 fL) vs. 10.9 fL, (9.4–12.3 fL), *p* = 0.046) were higher than in normo-prolactinaemic patients.

### 3.5. Associations between Bleeding Symptoms, Markers of Disease Severity and Hormones

During the acute PUUV infection, eight patients presented with mild bleeding symptoms. The bleeding symptoms associated with a more severe course of the infection, indicated by a longer hospital stay (median 8 days, range 7 to 22 days vs. median 6 days, range 3 to 10 days, *p* < 0.001), lower minimum systolic blood pressure (92 mmHg (60–119 mmHg) vs. 114 mmHg (80–136 mmHg), *p* = 0.003), and lower platelet counts (24 × 10^9^/L (4–97 × 10^9^/L) vs. 60 × 10^9^/L (21–389 × 10^9^/L), *p* = 0.002).

Bleeding symptoms were more frequent among patients with peripheral hormone defects than in those with normal hormone levels (67% vs. 0%, *p* = 0.001). The patients with bleeding symptoms had lower fT4 levels (11.6 pmol/L (4.8–14.1 pmol/L) vs. 14.3 pmol/L (6.4–17.5 pmol/L), *p* = 0.009) and higher cortisol levels (655 nmol/L (358–1454 pmol/L) vs. 401 pmol/L (180–655 pmol/L), *p* = 0.013) than the patients without bleeding symptoms.

## 4. Discussion

In the present study, hormonal defects were common during acute PUUV infection. Two thirds of the patients presented with overt (i.e., clinical) gonadal and thyroid axis defects. Male patients had more central and peripheral hormone defects than female ones. The acute peripheral hormone defects associated with a lower platelet count, more severe AKI and a longer hospital stay, compared to patients with normal hormone levels. Two thirds of the patients with peripheral hormone deficiencies had also mild bleeding symptoms. D-Dimer, a marker of thrombosis and fibrinolysis, was higher in patients with overt hypothyroidism, compared to patients with normal thyroid hormone levels. Six patients had central hormone deficits at the one-month follow-up visit; these hormonal deficiencies were not associated with the severity of acute PUUV infection.

The hormonal disturbances observed in this prospectively studied cohort were mostly in line with the hormone alterations reported in our previous, retrospective study [15]. Hypogonadism was even more prevalent in the present study, 80% of the male patients having overt hypogonadism, while half of the men were hypogonadal in the previous retrospective study [15]. LH levels were significantly lower in the acute phase than at the one-month control visit in the present study, reflecting a predominantly central origin of hypogonadism. Low levels of testosterone (and fT4) were related to leukocytosis in the acute phase. Low testosterone levels also associated with a prolonged hospital stay. Similarly, the degree of gonadal axis suppression has been related to the severity of acute illness in a previous study on patients admitted to critical care units [22].

As expected, serum cortisol level correlated with the severity of acute illness (degree of AKI and leukocytosis, duration of hospital stay) [15]. During viral infections, several cytokines have been shown to activate the hypothalamic-pituitary-adrenal (HPA) axis, leading to an increased release of glucocorticoids from the adrenal cortex [23]. Impaired kidney function may also have contributed to the increased cortisol levels, as the conversion of cortisol to cortisone is reduced during impaired kidney function [24]. The increased HPA axis activity and cortisol levels, in turn, may suppress the activity of the gonadal and thyroid axes [25].

Despite the increased cortisol levels in general, some patients had abnormally low levels of cortisol during the acute illness. Although none of these patients required hydrocortisone substitution during the acute PUUV infection or shortly thereafter, we continue the follow-up of these patients in order to detect possible hypocortisolism or other clinical hormone deficiencies without delay.

Studies on inflammation and thyroid function have shown that cytokines, e.g., IL1 alpha and beta, IL6, TNF alpha, and interferon gamma, downregulate the thyroid hormone synthesis pathway, decreasing the secretion of T4 and triiodothyronine, T3 [26]. During acute illness, previously euthyroid patients may present with low levels of T4 and T3, and elevated levels of reverse T3, all of which gradually normalise as the acute illness subsides [27]. This state called ‘euthyroid sick syndrome’ or ‘nonthyroidal illness syndrome’ [26,27], may account for some of the thyroid axis abnormalities observed in the present study.

In the present study, the patients with overt peripheral hormone defects had lower platelet counts and higher IPF%, compared to patients with normal hormone levels. This implies that despite the decreased platelet count thrombopoiesis is active also in patients with overt hormone deficits during acute PUUV infection [7]. Increased D-dimer and F1 + 2 further support ongoing thrombin formation, fibrinolysis and consumption of platelets as the mechanism of thrombocytopenia.

In our study, the patients with hypothyroidism had higher D-dimer levels than euthyroid patients, and fT4 levels correlated inversely with IPF%, indicating active fibrinolysis and platelet formation in the hypothyroid patients. Furthermore, slightly lower fT4 levels were noted in the patients with mild bleeding symptoms than in those without any bleeding symptoms. Our observations are in line with previous reports regarding the associations between altered haemostasis and thyroid function [18,19,28].

The serum cortisol levels during acute PUUV infection correlated with the levels of D-dimer, prothrombin fragments F1+F2, and IPF%, and inversely with the minimum platelet count. Increased levels of clotting factors and abnormalities in fibrinolytic parameters have been reported in patients with Cushing’s syndrome [18,19,29,30]. The coagulation pathway has been reported to be hyperactive [30] and an increased incidence of both venous and arterial thrombotic events has been reported among patients with endogenous hypercortisolism [31,32].

In this study, the balance between thrombin formation and fibrinolysis remained beneficial for patients with overt hormonal defects, as no thrombotic or severe bleeding events occurred. Mild bleeding symptoms were observed in only one fifth of the patients, who suffered from a more severe course of PUUV infection and more peripheral hormone defects, as well as more severe hypercortisolism. A thrombotic tendency has previously been related to several hormonal abnormalities, especially hypothyroidism and hypercortisolism [18,19]. In addition, Swedish studies have shown an increased risk of venous thromboembolism [33], myocardial infarction, and stroke [34] in patients with HFRS. The hormonal alterations noted during acute PUUV infection, e.g., low fT4 and high cortisol levels might, in part, protect the patients from major bleeding complications by shifting the haemostatic balance to the prothrombotic direction.

In conclusion, hormonal deficiencies were common during acute PUUV infection. Patients with peripheral hormonal deficiencies had a lower platelet count and higher IPF%, compared to patients with normal hormone levels, but no severe bleeding complications were observed. Acute peripheral hormone deficits associated with more severe AKI and a longer hospital stay. Most but not all of the acute hormonal defects were normalised by the one-month follow-up visit. As chronic hormonal defects have been diagnosed a few years after PUUV infection in as many as one sixth of patients in previous studies [15,16], we will continue to follow up the patients for the possibility of developing chronic hormone deficiencies that may require hormonal replacement therapy.

## Figures and Tables

**Figure 1 viruses-13-01818-f001:**
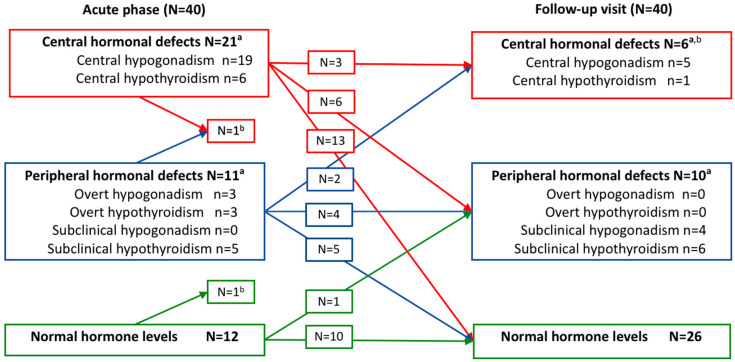
Central and peripheral hormone deficits in 40 patients during acute PUUV infection and at the one-month follow-up visit. ^a^ Eight patients had two different hormone defects in the acute phase, and two patients at the follow-up visit, respectively. ^b^ Hormone measurements were not available for two patients in the acute phase and two patients at the follow-up visit. N = Number of patients, n = number of hormonal defects.

**Figure 2 viruses-13-01818-f002:**
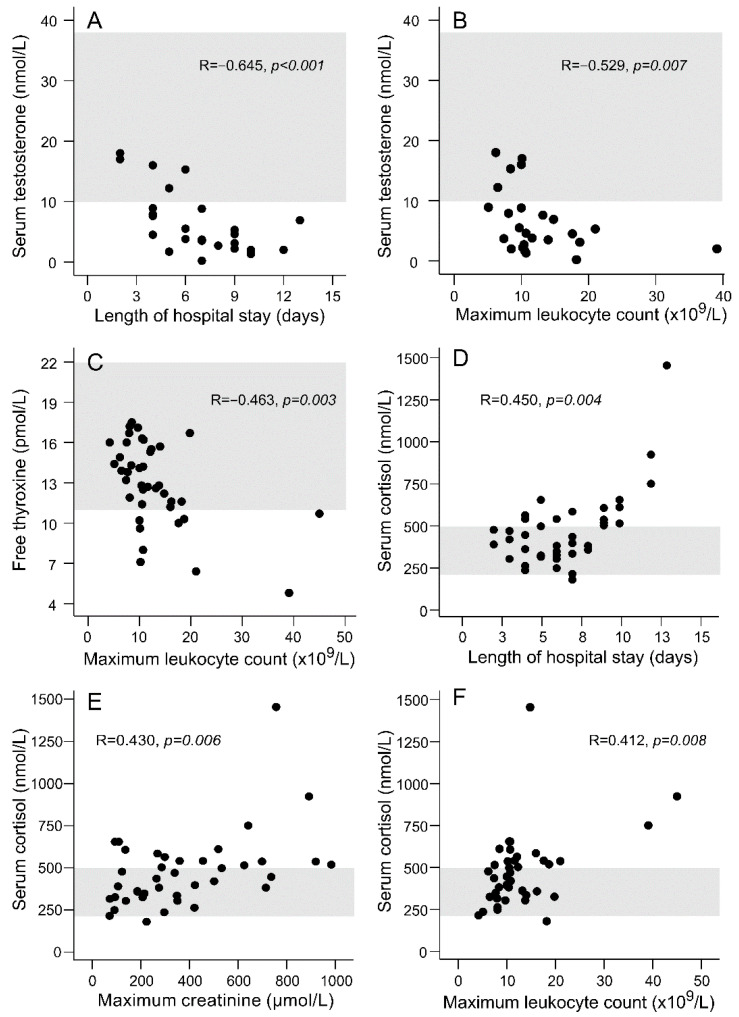
Scatter plots illustrating the correlations between serum testosterone level and the length of hospital stay (**A**) and maximum leukocyte count (**B**), between free thyroxine level and the maximum leukocyte count (**C**), between serum cortisol level and length of hospital stay (**D**), maximum creatinine level (**E**) and maximum leukocyte count (**F**) in patients with acute Puumala hantavirus infection. The reference range for each hormone is marked with shading.

**Figure 3 viruses-13-01818-f003:**
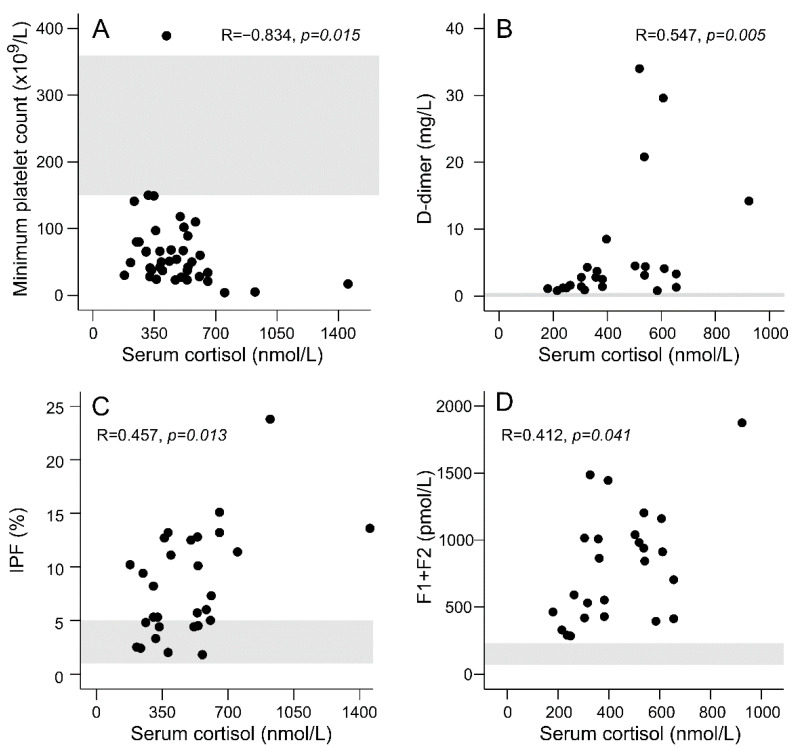
Scatter plots illustrating the correlations between serum cortisol level and the minimum platelet count (**A**), D-dimer (**B**), immature platelet fraction (IPF; **C**) and plasma prothrombin fragments (F1+F2; **D**) in patients with acute Puumala hantavirus infection. The reference ranges are marked with shading.

**Table 1 viruses-13-01818-t001:** Clinical characteristics and laboratory findings of 42 patients with acute Puumala hantavirus infection.

Clinical or Laboratory Variable	Median	Range
Age (years)	45	21–67
Sex (M/F)		27/15
BMI (kg/m^2^)	26.2	21–39
Length of hospital stay (days)	7	2–22
Weight change during hospital stay (kg)	−3.6	−0.3–(−12.5)
Min systolic BP (mmHg)	112	60–154
Min haematocrit	0.36	0.25–0.43
Max haematocrit	0.44	0.37–0.66
Min platelet count (×10^9^/L)	50	4–389
Max leukocyte count (×10^9^/L)	10.7	4.2–45.0
Max level of plasma CRP (mg/L)	68	16–244
Max plasma creatinine (µmol/L)	297	71–983
Plasma D-dimer (mg/L) *	2.8	0.6–34.0
Plasma prothrombin fragments (F1 + 2) (pmol/L) *	774	284–1875
Mean platelet volume (fL) *	11.0	9.4–13.1
IPF (%) *	7.3	1.8–23.8
Serum thrombopoietin (pg/mL) *	207	56–1258

Abbreviations: M = male, F = female, BMI = body mass index, BP = blood pressure, CRP = C-reactive protein, IPF = immature platelet fraction, min = minimum, max = maximum. Reference values: Haematocrit 0.39–0.50 for males and 0.35–0.46 for females, platelet count 150–360 × 10^9^/L, leukocyte count 3.4–8.2 × 10^9^/L, CRP <10 mg/L, creatinine 60–100 μmol/L for males and 50–90 μmol/L for females, D-dimer ≤ 0.5 mg/L, F1+F2 69–229 pmol/L, mean platelet volume 9–12 fL, and immature platelet fraction 1–5%. Serum thrombopoietin level depends on the platelet count. * Blood samples for the assessment of thrombin formation, fibrinolysis and thrombopoiesis were collected from 27 patients.

**Table 2 viruses-13-01818-t002:** Serum and plasma hormone levels during acute Puumala hantavirus infection and on a follow-up visit one month after discharge from hospital.

	Acute Phase (*n* = 40) **		Follow-Up Visit (*n* = 40) **		*p*-Value ^a^	Reference Range
**All patients**	Median	Range	Median	Range		
Cortisol (nmol/L)	441	180–1454	298	128–554	**<0.001**	170–500
ACTH (ng/L)	18.0	5.6–64.0	16.5	4.0–72.0	1.000	0.0–45.0
PRL (mU/L)	421	109–3415	205	98–551	**<0.001**	*
Free thyroxine (pmol/L)	13.0	4.8–17.5	15.0	9.8–19.8	**0.001**	11.0–22.0
Thyrotropin (mU/L)	3.1	0.57–9.10	2.20	0.33–6.30	**0.028**	0.27–4.2
TPO-Ab (kU/L)	13	4–108	13	5–103	0.656	<34
Growth hormone (GH) (ug/L)	0.90	0.1–21.5	0.3	0.03–8.00	0.220	*
IGF-1 (nmol/L)	13.0	2.9–29.0	23.0	13.0–37.0	**<0.001**	*
**Males (*n* = 25)**						
Testosterone (nmol/L)	4.6	0.2–18.0	13.6	6.7–29.5	**<0.001**	10.0–38.0
LH (U/L)	3.8	0.4–18.0	4.7	2.5–14.2	**0.027**	1.7–8.6
PRL (mU/L)	359.0	109–1283	186	98–533	**<0.001**	86–324
**Females (*n* = 15)**						
Oestradiol (nmol/L)	0.16	0.05–2.47	0.20	0.02–1.08	0.551	*
FSH (U/L)	4.2	1.4–63.4	6.4	3.4–69.6	**0.001**	*
PRL (mU/L)	471	313–3415	266	131–551	**0.001**	102–496

^a^ Comparison between the hormone levels measured in the acute phase and on the follow-up visit, Wilcoxon signed-rank test. * Oestradiol and FSH values were evaluated according to the phase of menstrual cycle and GH, IGF-1 and PRL values according to age or sex, as appropriate. ** Hormone measurements were not available in two patients in the acute phase and in two patients on the follow-up visit.

**Table 3 viruses-13-01818-t003:** Clinical findings during acute Puumala hantavirus infection in patients with normal hormone levels, and in those with overt central or peripheral hormonal defects.

	Normal Hormone Levels (*n* = 15)		Central Hormonal Defects ^a^ (*n* = 19)		Peripheral Hormonal Defects ^b^ (*n* = 6)		Over-All *p*-Value
	Median	Min–max	Median	Min–max	Median	Min–max	
Age (years)	45	21–63	46	28–67	38	25–67	0.741
Sex (M/F)	3/12		17/2		5/1		**0.001 ^c^**
BMI (kg/m^2^)	26	22–39	26	21–37	25	23–35	0.976
Length of HS (days)	6	2–10	6 ^#^	2–12	9 *	7–13	**0.020 ^c^**
Weight change during HS (kg)	−1.95	−0.3–(−12.5)	−5.6	−0.3–(−11.3)	−5.6	−3.2–(−7.0)	0.089
Min diuresis (ml/day)	1035	300–5100	805	150–2320	345	20–1200	0.143
Min systolic BP (mmHg)	108	80–139	110	60–154	117	86–125	0.939
Min platelet count (×10^9^/L)	67	27–389	42	4–141	30 **	5–66	**0.010 ^c^**
Max leukocyte count (×10^9^/L)	8.4	4.2–16.2	10.7	5.1–39.1	12.6	10.0–45.0	0.125
Max CRP (mg/L)	50	16–142	74	21–244	78	57–204	0.201
Max creatinine (μmol/L)	268	71–714	339	93–983	727 ***	274–919	**0.027 ^c^**

Abbreviations: M = male; F = female; BMI = body mass index; BP = blood pressure; CRP = C-reactive protein, HS = hospital stay, min = minimum, max = maximum. ^a^ Central hypogonadism or hypothyroidism, ^b^ Overt peripheral hypogonadism or hypothyroidism. ^c^ Statistically significant difference between the three groups, Kruskal–Wallis test for numerical variables, Fisher exact test for Sex.^#^ Patients with central hormone defects had a shorter hospital stay than patients with peripheral hormone defects (adjusted *p* = 0.042). Patients with peripheral hormone defects had * a longer hospital stay (adjusted *p* = 0.019), ** lower minimum platelet counts (adjusted *p* = 0.012) and *** higher maximum creatinine levels (adjusted *p* = 0.027), compared to patients with normal hormone values.

**Table 4 viruses-13-01818-t004:** Platelet count and markers of thrombopoiesis, thrombin formation and fibrinolysis during acute PUUV infection in patients with normal hormone levels, and in those with overt central or peripheral hormonal defects.

	Normal Hormone Levels (*n* = 15)			Central Hormonal Defects ^a^ (*n* = 19)			Peripheral Hormonal Defects ^b^ (*n* = 6)			Over-All *p*-Value
	n	Median	Min–max	n	Median	Min–max	n	Median	Min–max	
Min platelet count (×10^9^/L)	15	67	27–389	19	42	4–141	6	30 *	5–66	**0.010 ^c^**
MPV (fL) ^d^	9	10.8	9.5–12.3	13	11.4	9.4–13.1	4	12.5	10.9–12.8	0.059
IPF (%) ^d^	8	5.3	1.8–12.5	15	7.3	2.4–15.1	6	13.0 **	5.7–23.8	**0.013 ^c^**
Thrombopoietin (pg/mL) ^d^	9	128	59–347	11	151	56–648	6	268	162–1258	0.181
F1 + 2 (pmol/L) ^d^	8	429	284–1040	12	773.5	289–1487	5	1203	552–1875	0.088
D–Dimer (mg/L) ^d^	8	1.9	0.8–4.5	12	3.5	1.1–34.0	5	8.5	1.4–20.8	0.068

Abbreviations: MPV = mean platelet volume; IPF = immature platelet fraction %; F1 + 2 = plasma prothrombin fragments. Reference values: platelet count 150–360 × 10^9^/L, MPV 9.0–12.0 fL, IPF% 1.0–5.0%, F1+F2 69–229 pmol/L and D-dimer ≤ 0.5 mg/l. Serum thrombopoietin level depends on the platelet count. * The minimum platelet counts were significantly lower (adjusted *p* = 0.012) and ** the IPF% values higher (adjusted *p* = 0.010) in patients with peripheral hormone deficits than in those with normal hormone levels. ^a^ Central hypogonadism or hypothyroidism, ^b^ Overt peripheral hypogonadism or hypothyroidism. ^c^ Statistically significant difference between the three groups, Kruskal–Wallis test. ^d^ Blood samples for the assessment of thrombin formation, fibrinolysis and thrombopoiesis were collected from 25–29 patients.

## Data Availability

Original data are available as Appendix A.

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
