# Peer review of "Hormonal Defects Are Common during Puumala Hantavirus Infection and Associate with Disease Severity and Biomarkers of Altered Haemostasis"

_viruses, 2021, doi:10.3390/v13091818_

Round 1

Reviewer 1 Report

In this manuscript Tarvainen et al investigate correlations between coagulation and hormonal abnormalities in patients diagnosed with Puumalavirus infections. Clinical data were assessed during acute infections right after hospital admission and at follow-up visits, roughly one month after discharge.

The authors report clinical data from 42 patients, including 27 males and 15 females collected over the period of seven years in a Tampere University Hospital in Finland. Hormone defects were found to be more abundant in male than in female patients and significantly correlate with lower platelet counts, longer hospital stays and overall disease severity. Severe bleeding complications were however not observed in the patient cohort, independent of hormonal disbalances.  

The manuscript at hand represents a continuation of a previous clinical retrospective study from some of the authors. Their goal was to provide additional evidence for hormonal abnormalities during PUUV infections in a prospective setting that includes systematic follow-up data from convalescent patients one month after discharge. Of note, both their surveys found similar correlations, which underlines the relevance and validity of their results. The manuscript is well written and data are presented convincingly. The purely descriptive nature of the author’s observation slightly limits the overall impact of the presented data, which are nonetheless without any doubt, clinically and biologically relevant and worth publishing.

Overall acceptance of the manuscript for publication is highly recommended. However, minor changes to the paper are advisable. In general the manuscript would be more instructive if some of the data would be presented in bar graphs, scatter plots or by other visual presentation methods :       

Table 2: It may be helpful to display the changes as foldchange in bar charts. This may emphasize i.e. the extent of the Testosterone vs. Estradiol miss regulation in acute infections.

Lines 227-233: The authors should state what hormone concentrations they are referring to (during acute infection or follow-up visit). Moreover, the authors should include correlation plots to support and visualize their statements.

Chapter 3.4 and 3.5: Again, correlation plots should be utilized to display correlations or, if the authors deem it relevant the lack thereof.    

Author Response

RESPONSES TO REFEREE 1

In this manuscript Tarvainen et al investigate correlations between coagulation and hormonal abnormalities in patients diagnosed with Puumalavirus infections. Clinical data were assessed during acute infections right after hospital admission and at follow-up visits, roughly one month after discharge.

The authors report clinical data from 42 patients, including 27 males and 15 females collected over the period of seven years in a Tampere University Hospital in Finland. Hormone defects were found to be more abundant in male than in female patients and significantly correlate with lower platelet counts, longer hospital stays and overall disease severity. Severe bleeding complications were however not observed in the patient cohort, independent of hormonal disbalances.  

The manuscript at hand represents a continuation of a previous clinical retrospective study from some of the authors. Their goal was to provide additional evidence for hormonal abnormalities during PUUV infections in a prospective setting that includes systematic follow-up data from convalescent patients one month after discharge. Of note, both their surveys found similar correlations, which underlines the relevance and validity of their results. The manuscript is well written and data are presented convincingly. The purely descriptive nature of the author’s observation slightly limits the overall impact of the presented data, which are nonetheless without any doubt, clinically and biologically relevant and worth publishing.

In general the manuscript would be more instructive if some of the data would be presented in bar graphs, scatter plots or by other visual presentation methods.

We thank the reviewer for the overall positive feedback and appreciate the suggestions. We have now added Figure 2 with scatter plots illustrating the strongest correlations between the hormone levels and the markers of disease severity, and Figure 3, which shows the associations of cortisol levels with platelets and markers of thrombin formation and fibrinolysis. 

  • Table 2: It may be helpful to display the changes as foldchange in bar charts. This may emphasize i.e. the extent of the Testosterone vs. Estradiol miss regulation in acute infections.

We thank the reviewer for good suggestions for improving the presentation of the data. We find it important, however, to keep the original Table 2 in the manuscript, to show all the differences in plasma and serum hormone levels between the acute phase and the follow-up visit. We, however, added new scatter plot figures to show the correlations between hormone levels and markers of the clinical severity of PUUV infection (Figures 2 and 3). The manuscript is quite comprehensive and includes 14 pages, and therefore we restrained from adding the bar charts.

  • Lines 227-233: The authors should state what hormone concentrations they are referring to (during acute infection or follow-up visit). Moreover, the authors should include correlation plots to support and visualize their statements.

Thank You for this relevant comment. We have now revised the text as follows (lines 235-236): “There were several significant correlations between the hormone concentrations measured in the acute phase and the markers of PUUV infection severity.”

We have also added Figure 2 with scatter plots illustrating the strongest correlations between the hormone levels and the markers of disease severity.

  • Chapter 3.4 and 3.5: Again, correlation plots should be utilized to display correlations or, if the authors deem it relevant the lack thereof.    

Thank You for this suggestion. We have now added scatter plot figures to visualize the correlations between the serum cortisol level and the minimum platelet count, D-dimer, IPF and F1+F2, in Figure 3.

Reviewer 2 Report

The authors conducted a prospective study on hormones and the coagulation system in patients with PUUV infection. Hormonal abnormalities were found in the acute phase of PUUV infection, and some hormonal abnormalities were still present one month after discharge. There was also a correlation between the severity of PUUV infection and hormonal alterations. Overall, the report is well organized, but there is still room for improvement in the following points:

1.It would be better to define more clearly the definition of the severity of PUUV infection.

2. the description of the correlation between hormone levels and disease severity (3.3 and 3.5) could be more easily understood with figures and tables.

Author Response

RESPONSES TO REFEREE 2

The authors conducted a prospective study on hormones and the coagulation system in patients with PUUV infection. Hormonal abnormalities were found in the acute phase of PUUV infection, and some hormonal abnormalities were still present one month after discharge. There was also a correlation between the severity of PUUV infection and hormonal alterations. Overall, the report is well organized, but there is still room for improvement in the following points:

1) It would be better to define more clearly the definition of the severity of PUUV infection.

We thank the reviewer for the positive feedback and appreciate the suggestions.

There is no widely approved and exact definition for severe PUUV infection in scientific literature. The highest plasma creatinine concentration measured during the hospital stay is commonly used to illustrate the severity of PUUV-induced AKI. The difference between the maximum and minimum body weight during the hospital stay (weight change during hospitalization) reflects the disturbances in the fluid balance, and usually correlates with the severity of AKI and reflects the phenomenon of capillary leakage, as well. Blood leukocyte count and plasma CRP level reflect the degree of inflammation during the acute infection, and the lowest platelet count measured illustrates the patient’s risk for bleeding. The length of hospital stay reflects the overall severity of illness. The determinants of capillary leakage in PUUV infection are low blood pressure at hospital admission, presence of clinical shock, high hematocrit, and, as mentioned above, greater change in weight during the hospital stay.

We have added new text into the revised version of the manuscript, as follows (Methods, lines 103 to 109): The highest plasma creatinine concentration measured during the hospital stay reflects the severity of PUUV-induced AKI. The difference between the maximum and minimum body weight during the hospital stay (weight change) reflects the severity of AKI and the phenomenon of capillary leakage. Blood leukocyte count and plasma CRP level reflect the degree of inflammation during the acute infection, and the lowest platelet count measured illustrates the patient’s risk for bleeding. The length of hospital stay reflects the overall severity of illness.

2) The description of the correlation between hormone levels and disease severity (3.3 and 3.5) could be more easily understood with figures and tables.

Thank You for the appropriate suggestion. We added scatter plot figures (Fig. 2 and Fig.3)  to represent the correlations of hormone levels with the severity of PUUV infection and the bleeding/coagulation parameters.

Reviewer 3 Report

This manuscript by Tarvainen et al. examines hormonal irregularities during acute PUUV infection. They report that during infection, a high proportion of patients experience either central or peripheral hormone defects, which resolve in most patients, but are still present in some upon follow-up visit. The authors also were able to correlate hormone defects with other markers of disease severity in PUUV infection such as those for kidney injury, hospital stay, and thrombocytopenia. The study design and methods used are appropriate to attempt to answer this question and their findings support the idea that this is something that should be monitored in HFRS patients. It will be interesting to note moving forward whether hormonal defects play a role in more severe infections such as those caused by Hantaan or Dobrava virus, or even in New World hantavirus infections. The ties of the data to various aspects of the clotting cascade are also very interesting and could play a role in more severe HFRS cases, in which thrombotic complications are more pronounced and important. Future prospective studies in this respect are warranted. I have some questions and minor comments regarding the manuscript.

Line 81: In the description of patient characteristics in the methods, the authors list some concomitant diseases seen in certain patients. Are any of the conditions listed known to impact the levels of the hormones examined in the paper?

Line 122: For the statistical comparisons, the authors used a Kruskal-Wallis test, a non-parametric ANOVA, to compare the differences between the three groups in most tables for continuous data. They say that after significant test result, they ran a Mann-Whitney, presumably between each individual comparison to see which were statistically different. I am wondering why this was done, rather than just a Dunn’s post-test following the Kruskal-Wallis, which is designed for this purpose? Changing to a comparison of only two sets of data vs the original three changes the test being performed and the results. Additionally, in the tables that show these data being compared, such as in tables 3 and 4, it just lists Kruskal-Wallis (or Fisher exact test), but is this the p value that is reported? Or is it from the follow-up Mann-Whitney? It is not clear

For the results in Table 2, I suppose it would be expected that after clearing infection many of the hormone levels would revert back to normal levels for each individual. I am wondering though if there would be any sort of overcompensation effect seen in these patients following a long stretch of either lowered or elevated hormone levels? Also, the authors include a reference range in the table, which many of the markers examined fall within, even if they are significantly higher or lower than the follow-up levels. It would have been nice to see some age-matched healthy controls included for direct comparison rather than only reference values.

In the discussion, the ties of various hormones to the immune system is mentioned, and I am wondering if the authors have examined any measures of the immune response, other than the CBC done here? Many of these hormones can have significant effects on the immune phenotype, which could have implications for infection outcome. Perhaps this is a future or ongoing study.

One thing that could be added to the paper is a figure or figures which include some of the correlation analyses done, for example between certain hormones and CBC values, etc. with various ones that are correlated included in the manuscript, and others which are not added as supplemental. It would just put all of the important data in one place for readers to see.

Author Response

RESPONSES TO REFEREE 3

This manuscript by Tarvainen et al. examines hormonal irregularities during acute PUUV infection. They report that during infection, a high proportion of patients experience either central or peripheral hormone defects, which resolve in most patients, but are still present in some upon follow-up visit. The authors also were able to correlate hormone defects with other markers of disease severity in PUUV infection such as those for kidney injury, hospital stay, and thrombocytopenia. The study design and methods used are appropriate to attempt to answer this question and their findings support the idea that this is something that should be monitored in HFRS patients. It will be interesting to note moving forward whether hormonal defects play a role in more severe infections such as those caused by Hantaan or Dobrava virus, or even in New World hantavirus infections. The ties of the data to various aspects of the clotting cascade are also very interesting and could play a role in more severe HFRS cases, in which thrombotic complications are more pronounced and important. Future prospective studies in this respect are warranted. I have some questions and minor comments regarding the manuscript.

1) Line 81: In the description of patient characteristics in the methods, the authors list some concomitant diseases seen in certain patients. Are any of the conditions listed known to impact the levels of the hormones examined in the paper?

Thank You for this good question. As the concomitant diseases of the patients were treated appropriately, they were unlikely to have any significant impact on the hormone levels measured. For example, the hypothyroid patient was on adequate levothyroxine substitution, and the patient with coeliac disease was on a strict gluten-free diet during hospitalization. The patients with COPD or asthma had a mild disease, well controlled without inhaled corticosteroids.

As these diseases occurred in only one or two patients each, we did not find it necessary to discuss their eventual, unlikely effects on the hormone levels in the manuscript.

2) Line 122: For the statistical comparisons, the authors used a Kruskal-Wallis test, a non-parametric ANOVA, to compare the differences between the three groups in most tables for continuous data. They say that after significant test result, they ran a Mann-Whitney, presumably between each individual comparison to see which were statistically different. I am wondering why this was done, rather than just a Dunn’s post-test following the Kruskal-Wallis, which is designed for this purpose? Changing to a comparison of only two sets of data vs the original three changes the test being performed and the results. Additionally, in the tables that show these data being compared, such as in tables 3 and 4, it just lists Kruskal-Wallis (or Fisher exact test), but is this the p value that is reported? Or is it from the follow-up Mann-Whitney? It is not clear.

We consulted our statistician, who confirmed that the SPSS version 26 does not use the Dunn’s post-test following the Kruskal-Wallis test. To avoid any confusion regarding the significance levels, we removed the phrase ”(p<0.0167 in the post-hoc comparisons of 3 groups)” from the  Statistics chapter.

Continuous numerical data were analysed by the Kruskal-Wallis test, and after a statistically significant test result, post hoc comparisons between the groups were performed using the Mann-Whitney U test. In Tables 3 and 4 the over-all p-value represents the p-values of the Kruskal-Wallis tests. Statistically significant p-values are bolded. If the Kruskal-Wallis test result was statistically significant, we performed the Mann-Whitney post hoc tests, and the adjusted p-values of the M-W tests are shown in the footnote below the tables, lines 271-273 and 303-304. 

3) For the results in Table 2, I suppose it would be expected that after clearing infection many of the hormone levels would revert back to normal levels for each individual. I am wondering though if there would be any sort of overcompensation effect seen in these patients following a long stretch of either lowered or elevated hormone levels?

Thank you for raising this important issue. Central hypogonadism and hypothyroidism were the most common hormonal deficiencies during the acute PUUV infection. Some of these patients had LH or TSH levels above the reference level and testosterone or fT4 levels within the reference range at the one-month control visit, during the recovery phase. Further two years of continued follow-up will show, if these were transient, compensatory rises in LH or TSH levels, explained by short term central hormonal deficiencies or euthyroid sick states. The second option is that these patients may develop chronic peripheral hormonal deficiencies. We have continued to follow up these patients in order to find out, whether the patients are developing chronic hormone deficiencies, and we will discuss this question, when we have analysed the results of the long-term follow-up.

4) Also, the authors include a reference range in the table, which many of the markers examined fall within, even if they are significantly higher or lower than the follow-up levels. It would have been nice to see some age-matched healthy controls included for direct comparison rather than only reference values. 

It would have been interesting to have age-matched healthy controls, but we did not unfortunately have that kind of control groups. The reference ranges used in the manuscript, however, have been controlled with a local population sample, and age- and sex-specific reference ranges from Fimlab Laboratories, Tampere, Finland, were applied in the present study, as appropriate (https://fimlab.fi/briefly-in-english).

5) In the discussion, the ties of various hormones to the immune system is mentioned, and I am wondering if the authors have examined any measures of the immune response, other than the CBC done here? Many of these hormones can have significant effects on the immune phenotype, which could have implications for infection outcome. Perhaps this is a future or ongoing study.

In this study there were no other measurements of the immune response than those mentioned in the manuscript. However, in our previous, retrospective study we found a negative correlation between serum testosterone and plasma IL-6 levels (Makela S. et al Hormonal deficiencies during and after Puumala hantavirus infection. Eur. J. Clin. Microbiol. Infect. Dis. 2010).

6) One thing that could be added to the paper is a figure or figures which include some of the correlation analyses done, for example between certain hormones and CBC values, etc. with various ones that are correlated included in the manuscript, and others which are not added as supplemental. It would just put all of the important data in one place for readers to see.

Thank you for this relevant comment. We have now added Figure 2 with scatter plots illustrating the strongest correlations between the hormone levels and the markers of disease severity, and Figure 3, which shows the associations of serum cortisol levels with platelets and markers of thrombin formation and fibrinolysis. Both these figures include scatter plots representing the strongest correlations between hormone levels and leukocyte and/or platelet counts.